# Contextual Factors Influencing Self-Management of Adolescents and Youth Living with HIV: A Cross-Sectional Survey in Lesotho

**DOI:** 10.3390/ijerph20010238

**Published:** 2022-12-23

**Authors:** Mapaseka Chabalala Nagenda, Talitha Crowley

**Affiliations:** 1Department of Nursing, Faculty of Medicine and Health Sciences, Stellenbosch University, Cape Town 7535, South Africa; 2School of Nursing, Faculty of Community and Health Sciences, University of the Western Cape, Cape Town 7535, South Africa

**Keywords:** adolescents, contextual factors, HIV, self-management, youth

## Abstract

Background: HIV treatment outcomes of adolescents and youth living with HIV (AYLWH) are lagging. One way to improve outcomes is through supporting AYLWH to acquire self-management skills. Although self-management is associated with improved health outcomes, condition-specific, individual/family, and social/environmental contextual factors influence self-management. We aimed to describe factors influencing the self-management of AYLWH in Lesotho. Methods: A cross-sectional survey design was used. AYLWH (*n* = 183) aged 15–24 were conveniently sampled from two HIV treatment sites in Lesotho. Participants completed self-report questionnaires in English or Sesotho. Results: Participants (89.1% female) had high HIV self-management scores (mean 92.7%, SD 5.3%) that corresponded with treatment outcomes (98.9% adherent and 100% viral load < 1000 copies/mL). This might be attributed to condition-specific factors, including once-daily doses (100%) and longer duration of treatment (81.4% on ART for more than 10 years). Participants were older (median age 22), and the majority (61.7%) had stable living conditions. Individual strengths were associated with higher self-management scores (*p* < 0.01) and mental health problems with lower self-management scores (*p* < 0.05). Most (97.9%) were satisfied with their health care services. Conclusions: Uncomplicated treatment regimens, longer duration of treatment, stable living conditions, individual strengths, good mental health, and satisfaction with healthcare services have a positive influence on self-management.

## 1. Introduction

The proportion of adolescents and youth living with human immunodeficiency virus (HIV) globally is growing. In 2020, 410,000 young people between the ages of 10 and 24 were newly infected with HIV. Among these, 150,000 were adolescents between the ages of 10 and 19 [1]. The regions with the highest numbers of adolescents and youth living with HIV (AYLWH) are Sub-Saharan Africa and South Asia. Lesotho has the world’s second-highest HIV prevalence rate. Twenty-five percent (25%) of the country’s population, or one in every four people, has HIV [2].

HIV treatment goals and health outcomes of AYLWH are lagging. Global antiretroviral treatment (ART) coverage amongst adolescents aged 10–19 is only 54% [1]. AYLWH tends to have low HIV treatment adherence and retention in care rates [3,4]. Viral suppression rates amongst adolescents living with HIV aged 10–19 years vary between 27% and 89% [5]. Further to this, adolescents’ transition from pediatric to adult care impact their treatment outcomes. A systematic review found that retention rates following the transition in most studies were around 70% and were worse amongst adolescents with unsuppressed viral loads [6].

It is well known that AYLWH must negotiate complex developmental changes together with managing a chronic illness. Mental and behavioral health factors impact the treatment of AYLWH [7]. They experience individual, family, and environmental challenges that influence how they manage to live with a chronic illness, including treatment fatigue, having unstructured lives and negative household dynamics, school commitments, poor service delivery at healthcare facilities, and stigma and discrimination [8,9]. 

Self-management refers to the behaviors that individuals must perform daily to live well with one or more lifelong conditions [10]. Self-management skills and abilities lead to positive behaviors such as taking treatment, attending appointments, and refraining from risky behaviors; this, in turn, leads to better health outcomes and well-being [11]. Evidence has shown that self-management interventions improve the health-related outcomes of people living with HIV and AYLWH [11,12]. Chronic illnesses affect youth in many ways during their transition to adulthood and adult care. Supporting them in developing independence and self-management skills is, therefore, a key task of healthcare professionals [13].

Although better self-management skills are associated with good health, both self-management skills and outcomes are directly affected by several contextual risk and protective factors. Understanding these factors is key to developing interventions to support youth with self-management and improving outcomes [10]. These contextual factors can be related to (i) the condition, (ii) individual and family characteristics, and (iii) the physical and social environments [10,14]. Condition-specific factors include factors such as the medication regimen, frequency of medication, being ill or stable, and the illness trajectory—perinatally or behaviorally infected [14,15]. Individual and family characteristics that influence self-management include age [16], psychological strengths and difficulties [17], mental health [7], health literacy [18], and family structure and functioning [9]. Factors in the physical and social environment such as schools, peers, health care services, and the neighborhood influence self-management [17,19]. 

The Lesotho government introduced antiretroviral therapy (ART) as an important strategy to combat HIV with an emphasis on improving care for key populations affected by HIV, such as adolescents. Some facilities are implementing differentiated service delivery models where youth corners have been established to provide a range of adolescent-friendly services. Youth corners are health facilities that are entirely established for young people to provide integrated comprehensive care [20,21]. Adolescent-friendly services have been associated with higher rates of retention in care and viral suppression [22,23]. 

Despite the HIV prevention and management strategies implemented in Lesotho, AYLWH still has poor health outcomes, which include poor retention in care, low viral suppression, and a high mortality rate [20,24]. Contextual factors influencing HIV self-management have not yet been explored in Lesotho. 

This study aimed at describing the (i) condition-specific, (ii) individual/family, and (iii) physical/social environmental factors that influence the self-management of AYLWH in Lesotho.

## 2. Materials and Methods

### 2.1. Study Design 

A cross-sectional survey design was used. This design assisted the researchers to describe the associated factors at a fixed point in time. 

### 2.2. Setting

Two health facilities were purposively chosen due to the high numbers of AYLWH attending care at these facilities. One facility was a youth corner at a hospital in the central region (Maseru District) of Lesotho which offers HIV services, while the other was a health center which is about 15 km from the hospital. 

### 2.3. Population and Sampling

The study target population was AYLWH 15–25 years attending ART services at the two selected health facilities (N = 368). Recruitment took place over a period of three months from November 2020 to January 2021. The researcher and a field worker approached AYLWH who had scheduled appointments with the guidance of healthcare workers. Participants were conveniently sampled if they were aware of their HIV status and on ART. 

The minimum required sample size was 184 participants to test the null hypothesis that none of the predictors was associated with self-management. The predictors included the following: condition-specific factors (type of medication, trajectory, and condition stability); individual/family factors (age, behavioral strengths, behavioral difficulties, mental health, health literacy, family structure, and functioning); and physical/social environment factors (healthcare access, transport, and satisfaction with services).

Of the 184 participants recruited, 1 participant withdrew. The final sample size was 183; 93 participants were recruited at the hospital and 90 were recruited at the health center. A larger number was recruited from the hospital because it serves a larger population than the health center. 

### 2.4. Data Collection and Measures

The data were collected through a validated self-report questionnaire available in English and Sesotho. The translation process included a forward translation of the questionnaire, which was then double-checked and compared to the English version of the questionnaire. 

The questionnaire included the following sections:Demographic characteristics: This included questions about the young person, their family, and their health. These questions pertain to the individual (age and gender) and family factors (family structure and functioning).Health and treatment: This section contained questions about the young person’s health, including their health literacy, mental health, and adherence to ART. The responses were mostly measured on a nominal level. The three questions on mental health were combined to create a mental health score. These questions pertain to condition-specific factors.Health care services: This section included five questions about the adolescents’ satisfaction with healthcare services and transportation. These questions relate to the physical and environmental factors and were measured on a nominal level (yes/no).Self-management: This section included 44 items that measured self-management. The Adolescent HIV Self-Management Scale (AdHIVSM) has been developed in South Africa and has high validity and reliability [25]. The questions were measured on a Likert-type scale from ‘strongly disagree/never = 1’ to ‘strongly agree/always = 4’. Some items were negatively phrased and inversely coded.Strengths and difficulties: This section contained 25 questions measuring the strengths and difficulties using the Strengths and Difficulties Questionnaire (SDQ) [26,27]. The measurement scale includes options for ‘not true = 0’, ‘somewhat true = 1’, and ‘certainly true = 2’. Items were coded according to the guidelines provided by the SDQ.

The researcher and the fieldworker obtained the most recent routinely performed viral load tests from patient folders. 

Questionnaires took approximately 40 min to complete. COVID-19 protocols were followed. The researcher and fieldworker assisted participants were needed. Informed consent was obtained before administering the questionnaire. The Health Research Ethics Committee waived parental consent for youth 15 to 18 years whose parents could not come to the hospital or health center.

### 2.5. Pilot Study

A pilot study was conducted on 18 AYLWH before the main study in September and October 2021. Minor changes were made to the English and Sesotho questionnaires, and the data were not included in the main study.

### 2.6. Reliability and Validity

Content validity and reliability of the AdHIVSM and SDQ instruments have been assessed in previous studies [25,26]. 

Reliability was tested using Cronbach’s alpha. Cronbach’s alpha of the SDQ questionnaire previously yielded reliability coefficients of 0.65–0.85 [26,27], indicating acceptable internal reliability. For AdHIVSM, Cronbach’s alpha was 0.84 in the South African study, where it was developed amongst adolescents aged 13–17 [25]. However, in the present study, the reliability of the AdHIVSM total scale was not acceptable (Cronbach alpha 0.5). Similarly, the SDQ subscale had a Cronbach alpha of 0.6. 

Stepwise item analysis was performed on the AdHIVSM, and 24 items that lowered the reliability of the scale were removed. Principal component factor analysis on the remaining 20 items revealed three factors: self-efficacy (role/identity management); resilience/positive attitude (emotional management); and medical management. The Kaiser–Meyer–Olkin measure of sampling adequacy was adequate for factor analysis (0.804), and Bartlett’s Test of Sphericity was statistically significant (*p* < 0.001). The Scree Plot suggested retaining three factors that explained 48.9% of the total variance in the scale. These components resembled the medical, emotional, and social/role management components of self-management as originally described by Lorig and Holman (2003) [28] (refer to Table A1 for the principal component analysis results). 

A reliability analysis was performed on the 20 AdHIVSM items that yielded a Cronbach alpha of 0.804. These items were, therefore, used to calculate an SM score for the AYLWH in this study.

Since the SDQ has been used in several parts of the world, the subscales were retained, but the results should be interpreted with caution. It is noted in the literature that authors commented on difficulties in the translation and back-translation of the SDQ in African languages [27].

### 2.7. Data Analysis

The data from the completed questionnaires were entered into and analyzed using Statistical Package for the Social Sciences (SPSS) for Windows software, version 27. The demographic and contextual variables, as well as the level of self-management, were described using appropriate descriptive statistics such as frequencies, percentages, means, percentages, and standard deviations. Frequency and percentage distribution tables were used to depict data at the nominal and ordinal levels, such as gender and Likert-scale responses. There were very few missing data. We performed a complete case analysis, including only those cases with complete data in the regression model.

Multiple regression analysis was used to determine the association between the independent variables (condition-specific, physical, and social environmental, and individual and family) and the dependent variable (self-management) while controlling for other variables. Independent variables that showed a level of significance of *p* < 0.1 on individual regression analysis were entered into the final model. A significance level of *p* < 0.05 was used. To control for confounding and complete reporting, all variables were included in the final model, even if the *p*-value was *p* > 0.05. There were no issues with multicollinearity in the final model.

## 3. Results

### 3.1. Demographic Characteristics

The demographic characteristics of participants are depicted in Table 1. The sample was predominantly female (89.1%, *n* = 163). The median age was 22 (Interquartile range 4) and, when categorized, most participants (61.2%; *n* = 112), were between the ages of 21 and 24. Almost two-thirds of participants (61.7%, *n* = 113) lived with their caregiver for more than 10 years.

### 3.2. Health and Treatment

Almost two-thirds of the participants (62.8%, *n* = 115) indicated that they were infected through mother-to-child transmission (MTCT) (see Table 2). With regard to mental health, most participants (99% to 100%) had not experienced depression or anxiety in the past 12 months. 

Most participants were treatment-experienced with 81.4% (*n* = 149) on ART for more than 10 years. The large majority (98.9%, *n* = 181) indicated that they never missed their antiretroviral treatment (ARVs) in the past month and all the participants (100%) had viral loads of <1000 copies per mL according to their patient records.

### 3.3. Healthcare Services

Table 3 indicates that most participants (97.9%, *n* = 179) were satisfied or very satisfied with services and that they liked attending appointments. However, transportation problems sometimes prevented more than two-thirds of the participants (69.9%, *n* = 128) from attending appointments.

### 3.4. Self-Management

As seen in Table 4, most of the HIV self-management item mean scores were above 3.5, indicating high levels of self-management. Lower mean scores were found in relational components, for example, having regular contact with friends (mean 2.92) and asking the doctor or nurse questions if there is something they do not understand (mean 3.42). For the total scale, the mean score was 92.7% (SD 5.3%). The minimum score was 76% and the maximum was 100%.

### 3.5. Strengths and Difficulties

Table 5 indicates that most participants reported several strengths and few difficulties. Although the participants reported several strengths, only 73.2% (*n* = 134) indicated that they had one good friend. Almost five percent (4.9%; *n* = 9) of participants reported that they are nervous in new situations and that they easily lose confidence. The total difficulties score had a mean of 1.26 (SD 1.7), a minimum of 0, and a maximum of 11. The strengths score had a mean of 9.9 (SD 3.8), a minimum of 8, and a maximum of 10.

### 3.6. Factors Influencing Self-Management

When the factors were included in the multiple regression model, higher levels of self-management were predicted with the individual variable of disclosure (the age of disclosure being after the age of 12 compared to between 10 and 12) (See Table 6). However, the mean difference between the self-management scores of the groups was only 0.088 (*p* = 0.02). Only two participants reported having their status disclosed to them between 10 and 12; the rest were disclosed after the age of 12 (*n* = 181). 

Self-management increased by 0.032 units with every one-unit increase in the individual strengths score (*p* < 0.01) and decreased by 0.074 units with every one-unit increase in the mental health score (*p* < 0.05). The self-management scores increased with the condition-specific variable of the duration of treatment. Participants who were on treatment for 6 to 10 years and more than 10 years had significantly higher scores compared to those on treatment for 1 to 5 years (*p* = 0.02 and *p* = 0.01). 

The environmental variable of sometimes having transport problems compared to never having transport problems was associated with higher self-management, although the difference between groups was only 0.029 units. The finding of transport problems increasing scores is interesting and requires further exploration. The other variables lost significance in the regression model. The model predicted 29.3% (Adjusted R squared) of the variance observed in the self-management score. This means that there might still be several other factors that explain the variation in self-management of AYLWH that were not included in this model.

## 4. Discussion

We aimed to describe the condition-specific, individual, family, and social and environmental factors influencing the self-management of AYLWH in Lesotho. Our sample was predominantly female (89.1%) and comprised older youth, which limits the generalizability of the results. In Lesotho, fewer males are accessing HIV treatment (76.6%) compared to females (84%) [20]. 

With regard to their health, the majority were perinatally infected with HIV, were disclosed to after the age of 12, and had no comorbidities or mental health problems. The low prevalence of mental health problems is unusual for adolescents. It may be that cultural or other contextual factors play a role, and this should be explored further. Most were on treatment for more than 10 years and once-daily treatment doses. Condition-specific factors such as uncomplicated regimens and condition stability improve self-management [15,18]. In our study, a longer duration of treatment was associated with higher self-management. Iribarren et al. (2019) explain that better self-management among treatment-experienced persons can be attributed to acquiring self-management skills over time [29].

Although early disclosure is usually advised (around the age of 10), there was a high rate of disclosure after the age of 12 in this study. This must be explored further in this context; however, it did not appear to influence the participants’ self-management abilities. Furthermore, although almost none of the participants knew their CD4 counts, adherence and viral suppression rates were high. The ART guidelines of the Government of Lesotho (2016) [20] no longer recommend routine CD4 counts; instead, the viral road is recommended. This may explain the knowledge deficit of the participants regarding their CD4 count. 

We had to adapt the original self-management scale to improve the reliability and validity of this sample. Therefore, the required self-management skills of AYLWH and abilities may evolve from adolescence to early adulthood [16] and should be explored further. Interestingly, the final adapted scale resembled the medical, role, and emotional management domains described amongst adult chronic illness populations [28].

Self-management scores in this study were high, with the medical management scale, which included items such as knowledge of viral load, having the highest mean scores. The self-management scale with the lowest mean items scores was the resiliency scale related to emotional management. Participants had a lower mean score for having regular contact with friends. Positive relationships and resources such as family, peers, and friends facilitate AYWH engagement in care [30].

The high self-management scores in this study corresponded with the high adherence and viral suppression rates found in this study. Crowley et al. (2020) [17] also found high self-management scores amongst adolescents living with HIV aged 13–17 years in Cape Town, South Africa, that corresponded with treatment adherence and viral suppression (80.6% viral load < 50 copies/mL). Conversely, a large retrospective cohort study in South Africa found that only 47.5% of adolescents aged 10–19 had suppressed viral loads at the most recent test and younger adolescents (aged 10–14 years) were more likely to be fully viral suppressed (viral load < 50 copies/mL) compared to older adolescents (15–19 years) [31]. 

High self-management scores may be explained by individual factors such as age, although age was not associated with self-management in our study. Our sample included predominantly older adolescents with apparently stable family conditions. More than half completed grades 10–12. Older adolescents might be more prepared and ready for the transition into adult care, as they become increasingly independent and more capable of self-management [14]. Other individual factors that improve self-management include behavioral strengths and good mental health as reported by most participants in this study. Individual strengths and fewer stressful life events have been associated with resilience amongst adolescents living with HIV aged 13 to 17 [32]. However, mental health problems are prevalent amongst AYLWH and have been associated with poor treatment outcomes [7]. Similarly, in this study, mental health problems decreased self-management although the prevalence of such was low.

Satisfaction with services may improve self-management. Adolescents reported high satisfaction and liked attending the clinic, although some reported transport problems. Cluver et al. (2018) found that health system factors improving retention in the care of AYLWH included staff who are kind and have time for adolescents [19]. The satisfaction of the participants with the services received might be attributed to the establishment of adolescent health corners where adolescents are provided with age-specific health care [20]. Youth-friendly services that are accessible motivate AYLWH to attend follow-up care [21]. Adolescents in this study who experienced transport problems had higher self-management scores. This should be explored further, but it could be that adolescents plan for medication pick-ups by friends and family members as it is the practice in Lesotho, especially if they have good adherence and viral suppression. 

To our knowledge, this is the first study to explore self-management amongst AYLWH in Lesotho. Due to the use of convenience sampling, AYLWH not attending services were excluded, and thus, the results may not accurately represent all AYLWH. Our sample included predominantly females and older youth. There was limited variability in self-management scores and some of the independent variables. Only two facilities were included in the study, and these were facilities that provided adolescent-friendly services. These findings may, therefore, not apply to other settings where such services are not rendered. The findings might be generalizable to settings where adolescent-friendly services are provided to older, predominantly female AYLWH. Future studies should consider using stratified random sampling to ensure representation of gender, age, and level of adherence/clinic attendance. A comparison of settings that provide adolescent-friendly services and those that do not might assist in teasing out the effect of these services on self-management.

## 5. Conclusions

The results of this study provide further support that uncomplicated treatment regimens, longer duration of treatment, stable living conditions, individual strengths, good mental health, and satisfaction with health care services have a positive influence on self-management. The required chronic disease self-management skills and abilities of AYLWH may be context-dependent and evolve and should be explored further.

## Figures and Tables

**Table 1 ijerph-20-00238-t001:** Demographic characteristics of the participants.

Variable	Frequency (*n*)	Percentage (%)
**Site**		
Hospital	93	50.8
Health Centre	90	49.2
**Gender**		
Male	20	10.9
Female	163	89.1
Other	54	29.5
**Age**		
15–17 years	31	16.9
18–20 years	40	21.8
21–24 years	112	61.2
**School enrolment**		
Yes	102	55.7
No	81	44.3
**Completed educational grade**		
6–9	45	24.5
10–12	101	55.1
**How long have you lived with the person who looks after you?**		
Less than a year	3	1.6
1–5 years	46	25.1
6–10 years	21	11.5
More than 10 years	113	61.7
**How many times have you moved from the house in the past 5 years?**		
0	145	79.2
1	25	13.7
2	10	5.5
3 or more	3	1.5

**Table 2 ijerph-20-00238-t002:** Health and treatment information.

Variable	Frequency (*n*)	Percentage (%)
**How did you become infected with HIV?**		
At birth/from my mother (MTCT)—Yes	115	62.8
By having sex—Yes	62	33.9
Forced sex or abuse—Yes	13	7.1
**At what age did you find out that you were HIV-positive?**		
Between the ages of 10 and 12	2	1.1
After the age of 12	181	98.9
**Most probable route of infection (Researcher determined)**		
Perinatally	114	62.3
Behaviourally	69	37.7
**During the past 12 months, have you felt the following for 2 weeks in a row? Sad, angry, or depressed**		
No	182	99.5
Yes	1	0.5
**During the past 12 months, have you felt the following: You lost interest in most things that usually give you pleasure?**		
No	183	100
**During the past 12 months, have you felt the following: Worried or anxious most of the time?**		
No	182	99.5
Yes	1	0.5
**How long have you been taking your medication (ARVs)?**		
Less than 1 year	5	2.7
1–5 years	20	10.9
6–10 years	9	4.9
More than 10 years	149	81.4
**In general, over the past month, how often did you miss taking your ARVs?**		
I never miss any of my ARVs	181	98.9
I miss my ARVs a little bit of the time	2	1.1
**How many tablets do you take every day?**		
1	183	100
**What is your most recent CD4 count?**		
I don’t know	174	95.1
I know	9	4.9
**Other illnesses**		
Diabetes	2	1.1
No Illnesses	181	98.9

**Table 3 ijerph-20-00238-t003:** AYLWH perceptions of health care services.

Variable	Frequency (*n*)	Percentage (%)
**My health provider treats me with respect**		
Usually	57	31.1
Always	126	68.9
**In general, how satisfied are you with the services you receive at the clinic/hospital?**		
Very dissatisfied	4	2.2
Satisfied	68	37.2
Very satisfied	111	60.7
**Do you like going to the clinic/hospital?**		
Yes	183	100
**Do transport problems prevent you from going to the clinic/hospital?**		
Never	55	30.1
Sometimes	128	69.9

**Table 4 ijerph-20-00238-t004:** HIV self-management (adapted 20-item scale).

Domain	Item Wording	Mean (SD)
**Self-efficacy (Role/identity management)**	I can achieve as much as other people who don’t have HIV	3.90 (0.30)
	I am confident that I can take care of my health	3.96 (0.21)
	I would cope if I told someone about my HIV status and that person didn’t accept it or ignored me	3.97 (0.21)
	I decided by myself whom I want to tell about my HIV status	3.89 (0.38)
	I plan how to take my ARVs when I am not at home (for example, when I am out with friends or go on a school camp)	3.99 (0.74)
**Resiliency/positive attitude (Emotional management)**	I can cope with it if people say nasty or hurtful things about people living with HIV	3.55 (0.49)
	Doing things I like (for example listening to music, reading, or playing sports) helps me to cope	3.51 (0.56)
	I am to be independent (taking care of myself)	3.39 (0.58)
	I aim to enjoy life, feel good and have fun	3.73 (0.44)
	I do things to improve my health (for example, by exercising or eating healthy foods)	3.55 (0.52)
	I have regular contact with friends (for example, at school or in my community)	2.92 (0.73)
**Medical management**	I attend clinic appointments on scheduled dates (for example, I use a calendar, phone, or my clinic card to remind myself)	3.81 (0.39)
	I ask the doctor or nurse questions when there is anything I don’t understand	3.42 (0.62)
	I understand why I am taking ARVs	3.85 (0.36)
	I know the names of my ARVs	3.70 (0.58)
	I know at what times I should take my ARVs	3.81 (0.48)
	I know what to do when I miss the time to take my ARVs	3.84 (0.37)
	I understand what will happen if I don’t take my ARVs every day	3.84 (0.37)
	I know what my viral load is	3.77 (0.49)
	I know what my viral load should be	3.77 (0.49)

**Table 5 ijerph-20-00238-t005:** Strengths and difficulties.

Variable	Not True *n* (%)	Somewhat True *n* (%)	Certainly True *n* (%)
I try to be nice to other people. I care about their feelings.	0	0	183 (100)
I’m restless, I can’t stay still for long.	171 (93.4)	10 (5.5)	1 (0.5)
I get a lot of headaches, stomach aches, and other sicknesses.	172 (94.0)	10 (5.5)	1 (0.5)
I usually share with others (food, games, pens, etc).	1 (0.5)	2 (1.1)	180 (98.4)
I get very angry and often lose my temper.	182 (99.5)	1 (0.5)	0
I am usually on my own. I play alone or keep to myself.	182 (99.5)	1 (0.5)	0
I usually do as I am told.	0	0	183 (100)
I worry a lot.	177 (96.7)	6 (3.3)	0
I am helpful when someone is hurt, upset, or feeling ill.	0	0	183 (100)
I am fidgeting or squirming.	183 (100)	0	0
I have one good friend or more.	5 (2.7)	44 (24.0)	134 (73.2)
I fight a lot. I can make other people do what I want.	183 (100)	0	0
I am often unhappy, downhearted, or tearful.	182 (99.5)	1 (0.5)	0
Other people my age generally like me.	7 (3.8)	64 (35.0)	112 (61.2)
I am easily distracted. I find it difficult to concentrate.	180 (98.4)	3 (1.6)	0
I am nervous in my new situations. I easily lose confidence.	174 (95.1)	9 (4.9)	0
I am kind to younger children.	2 (1.1)	3 (1.6)	178 (97.3)
I am often accused of lying or cheating.	181 (98.9)	1 (0.5)	1 (0.5)
Other children or young people pick on me or bully me.	182 (99,5)	1 (0.5)	0
I often volunteer to help others (parents, teachers, children, etc.).	2 (1.1)	3 (1.6)	178 (97.3)
I think before I do things.	0	0	183 (100)
I take things that aren’t mine from home, school, or elsewhere.	181 (98.9)	1 (0.5)	1 (0.5)
I get on better with adults than with people my age.	148 (80.9)	28 (15.3)	7 (3.8)
I have many fears. I am easily scared.	182 (99.5)	1 (0.5)	0
I finish the work I am doing. My attention is good.	1 (0.5)	0	180 (98.4)

**Table 6 ijerph-20-00238-t006:** Logistic regression analysis of self-management score on predictor variables.

	Crude Coefficient			Adjusted Coefficient		
	Coef.	*p*-Value	95% CI	Coef.	*p*-Value	95% CI
**Site: Health Centre**	Reference			Reference		
**Hospital**	0.016	0.042	0.001 to 0.031	−0.008	0.31	−0.024 to 0.007
**Individual: family stability—times moved house in the past 5 years**	−0.009	0.078	0.018 to 0.001	0.001	0.24	−0.008 to 0.011
**Individual: Age of disclosure**						
**10 to 12 years**	Reference			Reference		
**>12 years**	0.084	0.025	0.011 to 0.158	0.088	0.02	0.016 to 0.159
**Individual: Health literacy—Don’t know CD4 count:**						
**Yes**	Reference			Reference		
**No**	−0.08	<0.001	−0.114 to −0.046	−0.015	0.43	−0.050 to 0.021
**Condition: trajectory**						
**Perinatal**	Reference			Reference		
**Behavioral**	−0.02	0.014	−0.04 to −0.004	0.003	0.74	−0.015 to 0.021
**Condition duration:**						
**less than 1 year**	Reference			Reference		
**1–5 years**	0.045	0.069	−0.004 to 0.094	0.040	0.08	−0.005 to 0.012
**6–10 years**	0.039	0.159	−0.015 to 0.093	0.069	0.02	0.027 to 0.093
**>10 years**	0.086	0.000	0.042 to 0.131	0.078	0.01	0.125 to 0.129
**Environment: Transport**						
**Never**	Reference			Reference		
**Sometimes**	0.0398	<0.001	0.024 to 0.056	0.029	<0.001	0.013 to 0.045
**Strengths score**	0.039	<0.001	0.020 to 0.059	0.032	0.001	0.013 to 0.052
**Difficulties score**	−0.011	<0.001	−0.015 to −0.006	−0.003	0.280	−0.008 to 0.002
**Mental health score**	−0.091	0.016	−0.164 to −0.017	−0.074	0.048	−0.147 to −0.001
**Constant**				0.652	<0.001	0.414 to 0.889

## Data Availability

The data that support the findings of this study are available from the corresponding author.

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
