# Peer review of "Contextual Factors Influencing Self-Management of Adolescents and Youth Living with HIV: A Cross-Sectional Survey in Lesotho"

_ijerph, 2022, doi:10.3390/ijerph20010238_

Round 1

Reviewer 1 Report

The aim of this study was to assess self-management in adolescents and youth (AYLWH) living with HIV in Lesotho. 183 AYLWH were included in a cross-sectional survey design of whom 89% were female. Treatment success was exceptionally high in this study with 100% viral load <1000 copies/mL and the authors found high self-management scores corresponding with treatment outcomes.

The manuscript is well-written and easy to comprehend.

I have a few comments.

1)    It should be stated in the abstract and made clearer under Limitations in the Discussion that the study included 89% females and predominantly older youths which limits the generalizability of the results. Perhaps a comment on how a future study could be designed to include more males and younger, given that 76,6% of males are accessing ART in Lesotho it is a little surprising that so few males were included.

2)    Only 0.5% state that they have felt sad/angry/depressed for two weeks in the past 12 months which is amazingly low for an adolescent population irrespective of HIV status. I recommend a comment regarding this in Discussion.

Author Response

Thank you for the comments.

1. Abstract and limitations made clear that there were predominantly females. Recommendations for future studies made.

2. A sentence on the prevalence of mental health issues was added in the discussion section.

3. Spell check performed.

Kind regards

Talitha

Reviewer 2 Report

Good article and very interesting, but we need to make some changes:
- Complete in: Study design "This project helped the researchers to describe the associated factors at a fixed point in time." - present UNIVARIATE logistics - in supplementary material; - in univariate and multivariate regression tables, it's necessary
to indicate the reference category for each variable included in
the model. - in the table of the final model leave only the variables with p<0.05

Author Response

Thank you for the comments.

The wording 'associated' was added to the study design.

  • Reference categories were added to the linear regression table.
  • Univariate/crude coefficients were added to Table 6.
  • We included all the variables in the final table to control for confounding.

Kind regards

Talitha

Reviewer 3 Report

The authors conducted a study of factors influencing self-management of youth living with HIV: A cross-sectional survey in Lesotho. I have a few recommended edits for clarity, mostly in the methods.

1. How was missing data handled?

2. I recommend including both adjusted and usadjusted estimates in your regression table.

3. Were there any issues with multicollinearity in your models?

4. Overall, the study should clarify what other settings these findings may be generalizable to.

5. Similarly, additional settings for future research should be recommended.

6. Double check for typos throughout.

Author Response

Thank you for the comments: 

1. There were few missing data and a complete case analysis was performed.

2. Unadjusted/crude coefficients added to Table 6.

3. There were no issues with multicollinearity in the final model.

4&5. Generalisability discussed as well as future research recommendations.

6. Typo's checked

Kind regards

Talitha

Round 2

Reviewer 2 Report

Alteração apenas no título da Tabela 6. "Análise de regressão linear" para "Análise de regressão logística" do escore de autogestão nas variáveis ​​preditoras.

Author Response

Thank you for the comment. We changed the title to: Logistic regression analysis of self-management score on predictor variables. of self-management score on predictor variables.

Reviewer 3 Report

The authors have effectively addressed my concerns.

Author Response

Thank you. We conducted another round of language editing and spell-checking.

Kind regards

Talitha